# Pain Level Classification from Speech Using GRU-Mixer Architecture with Log-Mel Spectrogram Features

**DOI:** 10.3390/diagnostics15182362

**Published:** 2025-09-17

**Authors:** Adi Alhudhaif

**Affiliations:** Department of Computer Science, College of Computer Engineering and Sciences in Al-kharj, Prince Sattam Bin Abdulaziz University, P.O. Box 151, Al-Kharj 11942, Saudi Arabia; a.alhudhaif@psau.edu.sa

**Keywords:** pain level determination, speech signals, GRU-Mixer, Log-Mel spectrograms

## Abstract

**Background/Objectives**: Automatic pain detection from speech signals holds strong promise for non-invasive and real-time assessment in clinical and caregiving settings, particularly for populations with limited capacity for self-report. **Methods:** In this study, we introduce a lightweight recurrent deep learning approach, namely the Gated Recurrent Unit (GRU)-Mixer model for pain level classification based on speech signals. The proposed model maps raw audio inputs into Log-Mel spectrogram features, which are passed through a stacked bidirectional GRU for modeling the spectral and temporal dynamics of vocal expressions. To extract compact utterance-level embeddings, an adaptive average pooling-based temporal mixing mechanism is applied over the GRU outputs, followed by a fully connected classification head alongside dropout regularization. This architecture is used for several supervised classification tasks, including binary (pain/non-pain), graded intensity (mild, moderate, severe), and thermal-state (cold/warm) classification. End-to-end training is done using speaker-independent splits and class-balanced loss to promote generalization and discourage bias. The provided audio inputs are normalized to a consistent 3-s window and resampled at 8 kHz for consistency and computational efficiency. **Results:** Experiments on the TAME Pain dataset showcase strong classification performance, achieving 83.86% accuracy for binary pain detection and as high as 75.36% for multiclass pain intensity classification. **Conclusions:** As the first deep learning based classification work on the TAME Pain dataset, this work introduces the GRU-Mixer as an effective benchmark architecture for future studies on speech-based pain recognition and affective computing.

## 1. Introduction

Pain evaluation is the identification and measurement of the level of pain a patient is feeling [1]. Although conventional measures such as the Numerical Rating Scale (NRS) or the Wong-Baker Faces Scale (WBFS) are available, these rely on the patient’s ability to communicate, which is not always feasible [2]. This is a severe limitation in infants, patients who are non-verbal, or those with cognitive impairment [3]. To overcome this, researchers have created objective and automatic solutions. Facial expression analysis, which evaluates brows, eye closure, or jaw tension using computer vision techniques to detect pain indicators, is a common approach [4]. Physiological measures like heart rate, skin conductance, and electromyography (EMG) have also been used because they are natural stress reactions of the body from pain [5]. Body posture and movement patterns are tracked in others to detect signs of distress, especially in individuals with chronic pain conditions [4].

In more recent times, speech has been another source that has been found promising, as voice changes tend to mirror distress related to pain. Compared to other modalities, speech-based pain assessment has some inherent advantages. It is completely non-invasive and easy to acquire with just a microphone and minimal apparatus. This makes it especially suited to clinical practice in emergency rooms or home care. Second, speech naturally conveys emotional and physiological information such as pitch, volume, or rhythmic changes that are typically automatic and outside of one’s control and therefore constitute good indices of internal states such as pain [6,7]. In contrast to facial expressions, which can be enacted or culture-conditioned, vocal reactions to pain will be more automatic and invariant across persons. In addition, speech can be tracked continuously throughout normal conversation, with passive and ongoing assessment being conducted without engendering any disturbance to the patient. As a result, speech-based systems have achieved highly successful results in detecting pain, even from individuals who cannot reliably self-report their symptoms.

For instance, Tsai et al. [3] developed a model for estimating patient pain levels during emergency triage based on vocal features and facial expressions from audio-video recordings. The study sought to address the shortcomings of self-reported pain scores, e.g., the NRS, in Taiwan’s triage system. Features were extracted from both modalities and classification models were trained to conduct binary and ternary pain intensity classification. While specific dataset information and splitting methods were not mentioned, the model performed with 72.3% accuracy for binary and 51.6% for ternary classification. Aung et al. [4] motivated the need for automatic systems capable of detecting pain-related expressions for chronic pain rehabilitation. To this end, the authors introduced the multimodal EmoPain dataset, which consists of high-resolution face videos, room and head-mounted audio, full-body 3D motion capture, and EMG signals of chronic pain subjects performing instructed and self-initiated physical actions. Pain intensity was annotated from facial expressions by eight raters, and four clinical practitioners labeled pain-related body behavior. Oshrat et al. [6] suggested a speech-based binary pain detection approach from prosodic features. Recordings were obtained from 27 neurologically injured patients in a clinical environment while they reported different pain intensities. Acoustic features were extracted with the Emobase 2010 set from OpenSMILE, in addition to two new feature types derived from RASTA-PLP coefficients. A support vector machine (SMO in WEKA) was trained and tested with five-fold cross-validation. The highest performance was 77.3% accuracy and Cohen’s kappa 0.47 with the combined feature set. Ren et al. [8] developed an automatic pain classification system from paralinguistic features of speech. Acoustic representations in the form of ComParE, MFCCs, and deep spectrum features served as input to Support Vector Machines (SVM) and Long Short-Term Memory (LSTM)-Recurrent Neural Networks (RNN). The dataset comprises 844 recordings obtained from 80 subjects during a cold pressor pain induction experiment. The dataset was split into speaker-independent training, development, and test sets for a three-class pain level classification problem. Performance was measured in terms of unweighted average recall, with the highest performance of 42.7% obtained using the LSTM-RNN model with deep spectrum features. Salinas et al. [9] explored speech-based pain detection as a more informative and culturally flexible alternative to existing assessment tools, such as the WBFS. These standard instruments are based on patients’ self-reporting, potentially subject to subjectivity and limitations across varied populations. For their experiment, 50 German speakers were subjected to thermal pain by submerging their hands in water from 40 °C to 47 °C while uttering sustained vowel sounds. Acoustic analysis revealed that higher pain intensities, particularly at 47 °C, were found to increase the mean fundamental frequency and acoustic energy, particularly for vowels associated with pain-related utterances. Gutierrez et al. [10] proposed a hybrid AI approach to pain detection based on the combination of facial gesture analysis and paralinguistic speech features. The system was designed to mitigate limitations of subjective pain assessment, particularly for patients with low communicative capabilities. Although a detailed dataset and split information was not provided, the model achieved 92% precision, 90% recall, and 95% specificity. Cascella et al. [11] proposed an AI-based framework for cancer pain assessment by fusing speech emotion recognition and facial expression analysis. A multilayer perceptron neural network was trained on 181 prosodic features using the EMOVO dataset, which contains simulated recordings of six basic emotions. Interview data of cancer patients were assessed for testing, including two case studies. Speech annotation and facial expression classification were manually conducted using ELAN software for the identification of pain-related cues. The emotion classification model achieved an accuracy of 84%, with high precision, recall, and F1-scores for emotion classes. Chang et al. [12] explored deep learning for infant cry recognition to bypass manual feature extraction by training neural networks directly on raw audio signals. They collected cry recordings from babies in a clinical setting, converted these into spectrogram images, and fed these into a 2D convolutional neural network. Specifics of the data, like the total number of recordings and the exact train-test split, were not provided. While the study itself did not mention classification accuracy or other common performance metrics, follow-up work citing this study has shown that CNNs based on similar spectrogram-based approaches can achieve around 78–82% accuracy at distinguishing pain cries from others. Salekin et al. [5] developed a deep learning approach to neonatal pain estimation by analyzing crying sounds. Pain-detection models were trained on a new Neonatal CNN (N CNN) and two pre-trained networks (VGG16, ResNet50), all applied to spectrogram images of audio recordings. Recordings were obtained from 31 neonates in a clinical NICU setting and annotated based on the Neonatal Infant Pain Scale by nurses. Spectrograms were generated from pain and non-pain cry segments, and data augmentation techniques were utilized. Evaluation was performed under a leave-one-subject-out (LOSO) cross-validation scheme. The N CNN and VGG16 models both achieved 96.77% accuracy with an AUC of 0.94, while ResNet50 achieved 83.87% accuracy and an AUC of 0.83. Thiam et al. [13] combined hand-engineered audio features and deep learning to make pain intensity predictions from breath sound recordings. Recordings were collected from the SenseEmotion dataset wherein paralinguistic audio was matched with induced pain. Manually designed and spectrogram-based deep features were both extracted from the audio, on which models for classification were trained. Using support vector machines and random forests on hand-designed features, leave-one-subject-out cross-validation produced 63.86% (SVM) and 82.61% (Random Forests) accuracy for para-linguistic data; Random Forest models of physiological modality also obtained 82.61% accuracy. Tsai et al. [7] employed an LSTM model using stacked bottleneck vocal features for emergency triage pain level classification. Prosodic and spectral features were first extracted and compressed into bottleneck representations. These were then fed into an LSTM network that was trained and tested on a subset of the emergency triage speech corpus with leave-one-subject-out cross-validation. The model achieved 72.3% accuracy for binary-class (pain or non-pain) and 54.2% accuracy for three-class (mild, moderate, severe) pain level classification.

This study introduces a novel speech-based approach that classifies vocalizations along three clinically relevant dimensions: pain versus no-pain, thermal valence as warm versus cold, and a pain-intensity dimension of mild, moderate, and severe, by coupling learnable Mel-spectrogram features with a lightweight bidirectional GRU-Mixer network. Figure 1 shows the schematic illustration of the proposed framework.

As seen in Figure 1, the pipeline proceeds via preprocessing, feature extraction, and classification: microphone waveforms are first resampled to 8 kHz, duration-normalized to a fixed three-second window via trimming or zero padding, and converted to 64-bin Log-scaled Mel-spectrograms that temper dynamic-range variation while retaining perceptually salient frequency cues. Each time-frequency map is then processed as a sequence whose 64-dimensional frames are passed through a three-layer bidirectional GRU; gated dynamics model contextual spectral trajectories, where after adaptive average pooling reduces the bidirectional activations to an utterance-level descriptor. A fully connected head with dropout regularization provides task-specific logits so that the same backbone can drive binary pain detection, binary cold–warm discrimination, and multiclass pain-intensity recognition with independent softmax outputs; all heads are optimized jointly under class-balanced cross-entropy losses with Adam optimization, L2 weight decay, and early stopping to prevent overfitting. Performance on stratified train–test splits is reported in accuracy, precision, recall, F1-score, and confusion matrix for all tasks.

While prior research has shown the promise of pain detection using speech, most of the current efforts are based on hand-tuned prosodic features or conventional classifiers, which limit their generalizability to heterogeneous populations and clinical environments. For instance, studies like Oshrat et al. [6] and Ren et al. [8] employed SVMs and conventional acoustic features and have moderate accuracy but lack flexibility in capturing the temporal dynamics of pain vocalizations. Moreover, certain papers release incomplete information about datasets or lack standardization in preprocessing that could restrict reproducibility and generalizability. In contrast, the proposed model in this study avoids such weaknesses with a completely trainable end-to-end framework based on Mel-spectrogram representations and a light-weight bidirectional GRU-Mixer backbone. This approach eliminates the requirement for manually constructed features, enables richer temporal modeling of speech, and supports multitask classification of presence, intensity, and thermal valence of pain in a single unified framework. With the addition of strong preprocessing and regularization methods, the system achieves higher adaptability and clinical practicability, especially in speaker-independent scenarios where traditional methodologies tend to fail.

The main contributions of this study are:(1)An end-to-end differentiable architecture integrating Mel-spectral analysis and GRU-Mixer temporal modeling, without hand-engineered prosodic features.(2)A speaker-independent evidence-based preprocessing algorithm that standardizes recordings but preserves clinically relevant acoustic patterns.(3)Complementary descriptions formed by integrating fixed spectral frames and non-stationary recurrent encodings to produce a rich joint spectral-temporal pain and thermal state signature.(4)A multi-task training framework whose regularization techniques and balanced metrics in combination enhance robust, real-time audio monitoring across pain presence, temperature sense, and graded pain intensity.(5)To the best of our knowledge, this is the first study to utilize the TAME Pain dataset for supervised classification tasks involving vocal expressions of pain. By applying an end-to-end deep learning architecture to this recently released corpus, we establish a benchmark for future work in speech-based pain recognition.

## 2. Materials and Methods

### 2.1. Preprocessing

Let x[n] denote the raw audio waveform sampled at an original sampling rate $f_{s}^{orig}$. The first step in preprocessing is to standardize this signal to a fixed form across the dataset [14]. This begins by resampling x[n] to a uniform target sampling rate $f_{s}=8000$ Hz. The resampled signal is given by;(1)$$x_{r}\left[m\right]=x\left[\left\lfloor \frac{m{\cdot f}_{s}^{orig}}{f_{s}}\right\rfloor \right],\;m=0,1,\ldots ,N_{r}-1$$
where $N_{r}$ is the number of samples in the resampled signal. Next, a temporal normalization step ensures that each waveform spans exactly 3 s. Given the fixed sampling rate, this corresponds to $N=3\cdot f_{s}=24,000$ samples. If the resampled waveform has fewer than 24,000 samples, it is padded with zeros at the end:(2)$$x_{p}\left[n\right]=\left\{\begin{array}{c} x_{r}\left[n\right],\;\;\;0\leq n<N_{r}\\ 0,\;\;\;\;\;\;\;\;\;N_{r}\leq n<N \end{array}\right.$$

If instead $N_{r}>N$, the waveform is truncated to the first 24,000 samples:(3)$$x_{p}\left[n\right]=x_{r}\left[n\right],\;\;\;0\leq n<N$$

The resulting signal $x_{p}\left[n\right]\in R^{24000}$ is a uniformly sampled, fixed-length waveform with consistent temporal resolution and dimensionality across all samples [15].

### 2.2. Feature Extraction with Mel Spectrogram

Given the preprocessed waveform $x\left[n\right]\in R^{24000}$, the next stage transforms the time-domain signal into a time-frequency representation known as the Mel spectrogram, which is designed to approximate the frequency sensitivity of the human auditory system [16]. The first operation is to segment the signal into overlapping frames for short-time analysis. Each frame consists of $N_{FFT}=1024$ samples, and the frames are spaced by a hop size of H=256 samples. Denote the frame index as *m* and the sample index within a frame as *n*. The *m*-th frame is defined as;(4)$$x_{m}\left[n\right]=x\left[n+mH\right]\cdot w\left[n\right],\;\;\;0\leq n<N_{FFT},$$
where *w*[*n*] is a Hann window function that tapers the edges of the frame to reduce spectral leakage. Each windowed frame is then transformed into the frequency domain using the Discrete Fourier Transform (DFT) [17]:(5)$$X_{m}\left[k\right]=\sum_{n=0}^{N_{FFT}-1}x_{m}[n]\cdot e^{-j2\pi kn/N_{FFT}},\;\;\;0\leq k<N_{FFT}$$

The magnitude spectrum is computed by taking the squared modulus [18]:(6)$$P_{m}\left[k\right]=\;{\left|X_{m}\left[k\right]\right|}^{2}$$

To project the linear frequency spectrum onto the Mel scale, a filter bank of *M* = 64 triangular filters $\left\{M_{j}[k]\right\}$ is applied to the power spectrum [19]. Each Mel filter $M_{j}[k]$ emphasizes a specific range of frequencies based on the Mel frequency formula:(7)$$Mel\left(f\right)=2595\cdot \;{Log}_{10}(1+\frac{f}{700})$$

The energy in the *j*-th Mel band for frame *m* is computed as;(8)$$S_{m}\left[j\right]=\sum_{n=0}^{N_{FFT}/2}M_{j}\left[k\right]\cdot P_{m}\left[k\right]$$

To compress the dynamic range and approximate human loudness perception, a logarithmic transformation is applied:(9)$$L_{m}\left[j\right]=\mathrm{Log}\left(S_{m}\left[j\right]+\epsilon \right),\;\epsilon \approx 10^{-9}$$

The result is the Log-Mel spectrogram, a matrix $L\in R^{T\times 64}$, where *T* is the number of time frames and 64 is the number of Mel frequency bands. Each row $I_{m}\in R^{64}$ represents the Mel-scaled spectral energy distribution at a specific time frame. This matrix serves as the input sequence for the temporal modeling stage that follows.

### 2.3. GRU-Mixer

The GRU-Mixer is a neural architecture designed to model sequential patterns in time-series data, in this case, Log-Mel spectrograms derived from speech, and aggregate them into a compact global representation for classification. It consists of stacked bidirectional GRU layers followed by temporal pooling and a fully connected classification head.

Let the input to the model be a sequence of Mel-spectral feature vectors:(10)$$x_{1},x_{2},\ldots ,x_{T}\in R^{F}$$
where *F* = 64 is the number of Mel bands and *T* is the number of time steps (i.e., spectrogram frames). The entire sequence can be represented as a matrix:(11)$$X=\left[x_{1};x_{2};\ldots ;x_{T}\right]\in R^{T\times F}$$

The GRU-Mixer first encodes the temporal dynamics using a stack of *L* = 3 bidirectional GRU layers [20]. A single GRU cell maintains a hidden state $h_{t}\in R^{H}$ and updates it at each time step according to the following gating equations:(12)$$z_{t}=\sigma \left(W_{z}x_{t}+U_{z}h_{t-1}+b_{z}\right)\;r_{t}=\sigma \left(W_{r}x_{t}+U_{r}h_{t-1}+b_{r}\right)\;{\tilde{h}}_{t}=tanh\left(W_{h}x_{t}+U_{h}({r_{t}⨀h}_{t-1})+b_{z}\right)\;h_{t}=\left(1-z_{t}\right)⨀h_{t-1}+z_{t}⨀{\tilde{h}}_{t}$$

In the bidirectional setting, there are two GRUs: one moving forward in time (from *t* = 1 to *T*) and one backward (from *t* = *T* to 1) [21]. Their outputs are concatenated:(13)$$h_{t}=\left[{\vec{h}}_{t};{\overset{\leftarrow }{h}}_{t}\right]\in R^{2H}$$

This allows the model to encode both past and future context at each time step. The output of the GRU block is a sequence [22]:(14)$$H=\left[h_{1};\ldots ,h_{T}\right]\in R^{T\times 2H}$$

Dropout is applied between layers to regularize the model and reduce overfitting [23]. Rather than using only the final GRU output or applying attention mechanisms, the GRU-Mixer performs temporal mixing by computing the mean of all GRU outputs across time:(15)$$\bar{h}=\frac{1}{T}\sum_{t=1}^{T}h_{t}\in R^{2H}$$

This global average pooling compresses the entire time sequence into a fixed-dimensional vector that summarizes the speaker’s vocal patterns over the entire utterance. The pooled vector $\bar{h}$ is then passed through a dropout layer and a linear transformation to produce class scores [24]:(16)$$o=W_{c}\bar{h}+b_{c}\in R^{2}$$

A softmax activation function is applied to obtain posterior probabilities over the two output classes [25];(17)$$\hat{y}=softmax(o)$$

The final prediction is made by selecting the class with the highest probability:(18)$$\hat{\gamma }=\arg\underset{i}{\max}{\hat{y}}_{i}$$

The model is trained using the cross-entropy loss function:(19)$$L=-\sum_{i=1}^{2}y_{i}Log({\hat{y}}_{i})$$
where *y*_*i*_ ∈ {0,1} is the true label in one-hot form. An ℓ_2_ regularization term is added via weight decay to prevent overfitting.

### 2.4. Dataset

TAME-Pain is a speech corpus collected from 51 young adults who underwent a cold-pressor test involving intermittent dips of each hand in very cold and comfortably warm water while reading brief sentences aloud [26]. Participants rated their pain on a one-to-ten scale every few sentences, assigning each utterance a ground-truth pain rating. The database comprises 7039 trimmed speech clips, about 5 h 11 min of speech, with an average clip length of 2.6 s and about 138 total utterances per speaker. A four-way aligned label is assigned to each clip: the participant’s self-reported pain rating, a pain/non-pain label, a three-class pain intensity (mild, moderate, severe), and the thermal condition (cold or warm). Participants were mostly in their early twenties (mean age ≈ 21 years, SD ≈ 4) and comprised 26 women, 22 men, and 3 non-binary participants with diverse racial and ethnic backgrounds. Audio was recorded with a close-talk microphone in a controlled setting, and each clip includes manual markers of background noise, speech errors, breaths, and cut-offs along with a quality rating from 0 (clean) to 4 (highly corrupted). Interestingly, 4658 clips were pain-labeled as clean, and only five files did not have pain labels. Additional metadata files note demographics and recording settings, so that TAME-Pain is realistic and well-structured for research on pain detection, severity classification, thermal state recognition, and resistant speech analysis in noisy settings. Figure 2, Figure 3 and Figure 4 show sample raw speech waveforms for the pain vs. the non-pain classes, cold and warm conditions, and multiclass pain level classification cases. While the x–axis of these figures shows the time in samples, the y–axis of these figures shows the amplitude.

## 3. Experimental Works and Results

The proposed methodology was implemented with Python 3.11.13 using PyTorch 2.2.2 and executed on a computer with an NVIDIA GeForce RTX 3090 GPU (24 GB memory). A holdout cross-validation method was used, where 80% of the speech dataset was used for training and 20% for testing. During the experiment, audio data sampled at 8 kHz was processed into 3-s clips (24,000 samples) and converted into Log-scaled Mel Spectrograms using 64 Mel bands, with an FFT window size of 1024 and a hop length of 256. The developed GRU-Mixer model includes a 3-layer bidirectional GRU with a hidden size of 64, adaptive average pooling, a dropout layer with a 0.5 dropout probability, and a last linear layer for binary classification. Training was carried out using the Adam optimizer, where a learning rate of 1 × 10^−4^ and a weight decay of 1 × 10^−5^ were considered. Mini-batches of size 8 were used for training and testing, and training was done for at most 20 epochs with early stopping activated after three epochs with no test accuracy increase [27]. The model was evaluated by testing with accuracy and loss metrics tracked per epoch, and the best-performing model, as measured by test accuracy, was preserved for final assessment. As mentioned earlier, three independent experimental works were carried out to validate the proposed method’s efficiency.

### 3.1. Classification of Pain and Non-Pain Scenarios

The experiments were initially carried out on the classification of pain and non-pain classes. As observed, this is an example of binary classification, and Figure 5 illustrates the training and test accuracies vs. epochs for the proposed GRU-Mixer-based method in the binary classification problem between pain and non-pain speech samples. The training accuracy rises steadily, passing over 88% at epoch 16, meaning the model is learning from the training data effectively. The test accuracy rises steadily in the initial epochs, reaching a peak of approximately 84% at around epoch 13, but with small fluctuations in the latter part. The gap between the training and test accuracy, especially after epoch 10, may be a sign of mild overfitting.

Figure 6 shows the training and test loss curves over epochs for the GRU-Mixer-based pain classification model. The training loss demonstrates a steady and consistent decline, dropping from approximately 0.64 to below 0.29 by epoch 16, indicating effective learning and convergence on the training data. In contrast, the test loss shows more fluctuation throughout training, with periods of increase after initial improvement. This variability in test loss, especially after epoch 6, suggests potential overfitting as the model continues to improve on the training set while generalization to unseen data becomes less stable.

Figure 7 shows the confusion matrix for the binary classification task of distinguishing between pain and non-pain speech segments using the proposed GRU-Mixer model. Out of all samples, the model correctly classified 740 non-pain instances and 424 pain instances, while misclassifying 94 non-pain samples as pain and 130 pain samples as non-pain.

Table 1 presents the performance evaluation scores for the classification of pain and non-pain conditions using precision, recall, and F1-score metrics. The model demonstrated higher performance in identifying non-pain instances, achieving a precision of 0.8506, a recall of 0.8873, and an F1-score of 0.8685. In contrast, pain instances were detected with slightly lower values, with a precision of 0.8185, a recall of 0.7653, and an F1-score of 0.7910. The macro-average scores, which treat both classes equally regardless of size, were slightly lower than the weighted averages, reflecting the imbalance between classes. Overall, the classifier achieved an accuracy of 83.86%, indicating a solid performance in distinguishing between pain and non-pain speech signals.

### 3.2. Classification of Cold and Warm Condition Classes

The second experiment was carried out on the classification of cold and warm condition classes. Figure 8 demonstrates the training and test accuracies of the proposed speech-based classification system for distinguishing cold versus warm thermal valence over 20 epochs. The model shows a consistent upward trend in both training and test performance, reaching over 90% accuracy on the training set and approximately 87% on the test set by the final epoch. The relatively narrow gap between the two curves suggests strong generalization capability and indicates that overfitting has been effectively mitigated, likely due to the inclusion of dropout regularization, early stopping, and balanced class weighting.

Figure 9 presents the training and test loss trajectories over epochs for the cold vs. warm classification task using the proposed GRU-Mixer-based architecture. As shown, both losses decrease substantially during the initial training stages, indicating effective learning. The training loss continues to decline steadily, while the test loss exhibits slight fluctuations after epoch 5, likely reflecting small variations in batch generalization rather than overfitting.

Figure 10 illustrates the confusion matrix for the binary classification task distinguishing between cold and warm thermal states based on vocal input. The model correctly classified 534 out of 658 cold samples and 689 out of 751 warm samples, yielding high sensitivity for both classes. Notably, the number of false positives (124 cold predicted as warm) and false negatives (62 warm predicted as cold) is relatively low, indicating a balanced performance without significant class bias.

Table 2 presents detailed performance metrics for the cold vs. warm classification task, highlighting the effectiveness of the proposed method across multiple evaluation dimensions. The model achieves a precision of 0.8960 for cold samples and a recall of 0.9174 for warm samples, reflecting its strong capability to both correctly identify cold utterances and reliably detect warm ones. The F1-scores for both classes, 0.8517 (cold) and 0.8811 (warm), indicate a well-balanced trade-off between precision and recall. The macro- and weighted average scores further support the model’s robustness, with macro-averaged F1 at 0.8664 and overall accuracy at 86.8%, suggesting consistent performance across class distributions.

### 3.3. Classification of Multiclass Pain Classes

The third experimental work was carried out on multiclass pain level classification. Figure 11 depicts the training and test accuracy curves over 16 epochs for the proposed method applied to multiclass pain level classification (mild, moderate, severe). The model exhibits a steady improvement in both training and test accuracy throughout training, ultimately reaching approximately 77.8% on the training set and 75.2% on the test set. The relatively close alignment between the two curves suggests effective generalization and minimal overfitting, likely due to the use of dropout and early stopping in the training protocol.

Figure 12 shows the training and test loss trajectories across 16 epochs for multiclass pain classification using the proposed GRU-Mixer model. Both loss curves decrease notably during early epochs, indicating effective initial convergence. The training loss continues to decline steadily, reaching approximately 0.53 by the final epoch, while the test loss exhibits some fluctuation between epochs 7 and 13. However, these variations remain within a relatively narrow range and do not indicate persistent divergence, suggesting that the model maintains generalization capacity. Oscillations in loss curves reflect the trade-off between learning genuine pain-related acoustic patterns in the model and overfitting training data idiosyncrasies, scaled by dataset size, diversity, and stochastic training dynamics.

Figure 13 presents the confusion matrix for the three-class pain intensity classification task based on vocal input. The model achieves high accuracy in identifying mild pain, correctly classifying 757 out of 834 samples. While performance on moderate and severe classes is comparatively lower, the model still correctly identifies 189 out of 334 moderate cases and 100 out of 220 severe cases. Misclassifications are most prominent between adjacent categories, with moderate samples often confused as mild (112 instances) and severe samples as moderate (64 instances), reflecting the inherent difficulty in differentiating between closely graded pain levels based solely on speech.

Table 3 provides a class-wise breakdown of precision, recall, and F1-scores for the multiclass pain classification task involving mild, moderate, and severe pain levels. The model performs strongest on the mild class, achieving a high precision of 0.8184, a recall of 0.9077, and an F1-score of 0.8607, which aligns with the high correct prediction count seen in the confusion matrix. In contrast, performance decreases for the moderate and severe classes, with moderate pain showing an F1-score of 0.5833 and severe pain reaching 0.5420, indicating the increased challenge in distinguishing these pain levels based solely on speech. The macro-averaged F1-score is 0.6620, highlighting the effect of class imbalance, while the weighted average F1-score is 0.7435. Overall, the model achieves an accuracy of 75.36 percent.

### 3.4. Ablation Study

In order to assess the individual and combined value of key training components in the proposed GRU-Mixer-based model for binary pain classification, a systematic ablation study was carried out. The target task was the discrimination of vocalizations labeled as pain or non-pain, and the investigation was focused on three specific elements of the training process, such as dropout regularization, L2 weight decay, and early stopping. Six test configurations were defined. The full model, which served as the baseline, had all three components. The remaining five variants systematically omitted one or more of these to ascertain their contributions. They were: (1) a model without dropout, (2) a model without weight decay, (3) a model trained without early stopping, (4) a model without both dropout and weight decay, and (5) a stripped-down setup without any of the three mechanisms. Each model was trained from scratch using the same GRU-Mixer architecture, same random seed, learning rate, batch size, and number of epochs. Stratified splitting was employed to preserve class balance between train and test sets, and Mel-spectrogram representations of the audio data were employed in all experiments. Performance metrics such as accuracy, precision, recall, and F1-score were computed on the test set for each variant to enable quantitative comparisons and also to ascertain the contribution of each component in effective pain detection from speech. Table 4 shows the results of the ablation study.

As seen in Table 4, removing individual components from the full GRU-Mixer model produced significant fluctuations in classification accuracy. The full model with dropout, weight decay, and early stopping achieved an accuracy of 82.71% and an F1-score of 76.83%. After dropout was dropped, accuracy dropped a bit to 81.70%, and F1-score dropped to 74.55%. Weight decay removal dropped performance further, to 81.27% accuracy and the lowest F1-score of all configurations at 73.42%. Surprisingly, disabling early stopping boosted recall up to 80.14%, its best of all settings, yet reduced precision (79.57%) and moderately raised F1-score (79.86%) compared to the complete model. The setting without both dropout and weight decay worked best overall in recall at 88.99% but at the cost of lesser precision (71.87%) and comparable F1-score (79.52%). The GRU-only model without any of the regularization methods surprisingly had the best accuracy at 84.22% and best F1-score at 78.55%, which indicates that the inherent GRU architecture itself is extremely powerful for the task. But the combined application of the regularization techniques in the whole model generated a fairer trade-off between all measures, suggesting its robustness in generalization.

## 4. Discussions

In this study, a speech-based deep learning framework was developed to classify vocal expressions of pain using a learnable temporal modeling architecture. The proposed method extracts time-frequency features from raw speech through Log-Mel spectrogram representations, which are then encoded using a stack of bidirectional GRU layers designed to capture contextual acoustic patterns over time. These recurrent components are integrated with temporal pooling and a fully connected classification head to form a compact, discriminative feature embedding. Unlike traditional approaches that rely on hand-engineered prosodic or spectral features, the system is trained end-to-end, allowing the network to learn task-relevant temporal dynamics directly from data. Each audio input is standardized to a fixed 3-s window and resampled to 8000 Hz, enabling consistent input across the dataset while preserving critical acoustic cues. Classification is performed using a supervised deep learning setup targeting binary discrimination between pain and non-pain conditions. In this setting, the model achieved a test accuracy of 86.80%, along with a macro-averaged precision of 87.17%, recall of 86.45%, and F1-score of 86.64%, indicating strong performance in capturing vocal correlates of pain. These results demonstrate the efficacy of the GRU-Mixer architecture for speech-based pain assessment and support its suitability for non-invasive, real-time applications in clinical and caregiving environments.

The choice of the GRU-Mixer model might at first glance look as much a matter of random selection of a neural model for the task of pain recognition. The rationale originates in the particular strengths of recurrent networks for representing sequential information such as speech. Bidirectional GRU layers allow the model to capture both past and future temporal relationships in vocal signals, especially relevant for finding subtle prosodic and spectral content changes associated with pain. Incorporating temporal pooling also allows learning compact global representations without depending on handcrafted features. So, although the GRU-Mixer is quite lean compared to more complex structures, it is an informed compromise between expressiveness and computational expense, and therefore well-suited for real-time or low-resource clinical settings.

To the best of our knowledge, there are currently no publicly available classification results, neither binary (pain vs. non-pain) nor multiclass (e.g., mild, moderate, severe), reported using the full TAME Pain dataset. While the dataset has been introduced in recent publications [26,28], prior work has focused primarily on small-scale pilot studies involving a limited number of participants, where conventional machine learning methods were applied. However, these early experiments do not provide any generalizable benchmarks for the broader dataset, which includes a much larger and more diverse sample. In this context, the approach presented in this study represents one of the first end-to-end deep learning applications on the TAME Pain dataset.

To evaluate the efficiency of our proposed model, we did our best to compare it against several commonly used machine learning approaches on multiclass pain level classification. For this purpose, we included Logistic Regression (LogReg), Support Vector Machines (SVM), k-Nearest Neighbour with k set to 5 (kNN-5), Decision Tree (DT), and Random Forest (RF). Each method was configured with suitable parameters to ensure a fair comparison. LogReg was trained with a maximum of 2000 iterations and balanced class weights. SVM with linear kernel used C = 2.0, gamma set to “scale”, probability estimates enabled, and class weighting. kNN-5 was trained with k = 5. DT was trained with class weighting, while RF used 400 estimators and balanced subsample weighting. All models were trained on an 80/20 stratified train-test split to preserve class distribution, and their performance was assessed using macro-precision, macro-recall, and macro-F1-score, respectively.

Table 5 presents a comparison between the proposed GRU-Mixer model and several classical machine learning algorithms with macro-precision, recall, and F1-score metrics, respectively. LogReg and DT received relatively lower scores, i.e., poor ability for maintaining performance on all the classes. SVM performed better overall, but more precisely in recall. The kNN-5 model possessed stronger precision but was marred by poor recall and hence a weaker F1-score. RF possessed the strongest precision among the traditional models but possessed weaker recall, which compromised its overall F1 performance. However, the proposed GRU-Mixer model provided a more balanced outcome, achieving higher macro-recall than most baselines and a competitive macro-precision, leading to the best macro-F1-score across all methods.

The proposed method has several important advantages over standard speech-based pain identification methods. Firstly, it employs an end-to-end learnable architecture, which does not involve handcrafting prosodic or spectral features, which can be biased and suboptimal concerning subjects and recording environments. By using Log-Mel spectrogram representations with bidirectional GRU layers, the model can identify fine-grained spectral patterns as well as long-term temporal dependencies present in vocal pain expressions. Two, the architecture is lean and computationally cheap, which enables the model to be utilized in real-time or resource-constrained environments such as mobile health applications or clinical monitoring devices. Third, the strategy employed is amenable to multitask classification (i.e., pain detection, intensity grading, thermal state) in a shared framework that enables more complete coverage of clinically useful dimensions with a single input modality. Finally, the speaker-independent training and testing protocols employed enhance model generalizability to unseen speakers, a necessary condition for realistic usability across diverse patient populations.

Despite the positive results achieved, this work has a series of limitations that are to be acknowledged. Firstly, while the GRU-Mixer model was excellent in binary classification, its performance in more sensitive tasks, such as multiclass pain intensity detection, was not so accurate, particularly for moderate pain, an intermediate class. This calls for subtler modeling or more sophisticated features to capture subtle acoustic differences. Second, the system is now purely based on acoustic features, excluding linguistic, contextual, or multimodal information (e.g., expressions or physiological signals), which could potentially enhance performance at more realistic real-world tasks. Third, although speaker-independent splits were used to ensure generalizability, demographic and linguistic variation within the dataset may again limit how much generalization the model can perform to underrepresented speakers or dialects. Finally, while using fixed-length input windows makes model training more convenient, it may not necessarily capture the dynamic nature of spontaneous speech in which pain expressions can change between different durations and intensities over time.

The proposed GRU-Mixer approach is demonstrated to possess clear novelty compared to existing works because it does not rely on hand-designed prosodic or spectral features but instead leverages the power of an end-to-end, fully trainable model. The architecture allows the model to simultaneously process both temporal and spectral features of pain speech, as well as allowing multiple tasks such as binary pain detection, intensity estimation, and thermal condition classification. In this context, the framework offers a tunable alternative to traditional feature-based approaches.

In terms of effectiveness, reported performance is good, with around 84 percent accuracy for binary pain detection and 75 percent multiclass pain intensity classification. However, these are based only on the recently released TAME Pain dataset. Since there are no other published benchmarks yet for this corpus, and the previous research used to be based on smaller or less controlled datasets, it remains difficult to compare the performance of the proposed method to prior work directly. Extended experimental verification using various databases would help determine the system’s generalizability.

One must also point out that, although the model shows excellent performance with binary classification tasks, the accuracy when recognizing subtler classes, such as moderate and severe pain, is lower. Furthermore, the current system relies exclusively on acoustic cues and lacks linguistic or multimodal data, which could improve performance in more realistic scenarios. Lastly, despite speaker-independent splits, the demographic and linguistic diversity of the dataset can, at worst, constrain how well the model can generalize to populations that are different.

## 5. Conclusions and Future Work

In this work, we proposed a novel speech-based deep learning system for automatic pain classification founded on a learnable temporal modeling approach. The system combines Log-Mel spectrogram features and a stack of bidirectional GRU layers along with temporal pooling for end-to-end discriminative acoustic feature extraction. A key strength of our approach is that it does not rely on hand-engineered prosodic or spectral features, but instead learns to recognize relevant vocal patterns from raw speech input itself, making it more versatile across speakers and conditions. Use of a fixed-length preprocessing pipeline and shallow recurrent architecture keeps the model efficient and suitable for real-time home or clinical care uses. Additionally, the framework supports multitask classification, pain presence, pain intensity, and thermal state (cold-warm condition) from a single audio stream, allowing for more informative, context-aware interpretation of vocal pain expressions. With speaker-independent splits and stratified evaluation, our method also demonstrates great generalization performance, showing its readiness for application in real-world, non-invasive pain monitoring systems. In future work, we aim to explore the integration of other modalities, such as language content or physiological signals, and extend the system to more diverse populations and languages for general clinical application.

## Figures and Tables

**Figure 1 diagnostics-15-02362-f001:**
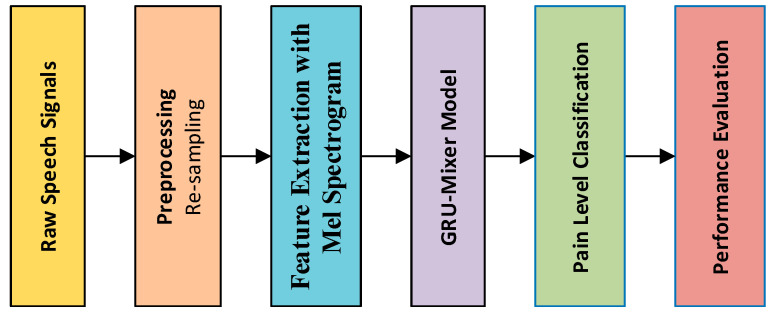
Schematic illustration of the proposed framework.

**Figure 2 diagnostics-15-02362-f002:**
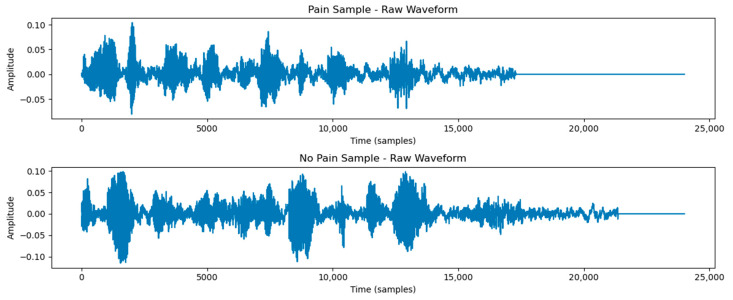
Raw speech waveforms for pain and non-pain samples.

**Figure 3 diagnostics-15-02362-f003:**
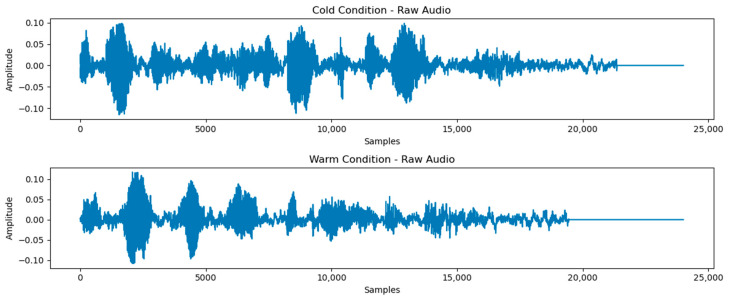
Raw speech waveforms for cold and warm conditions.

**Figure 4 diagnostics-15-02362-f004:**
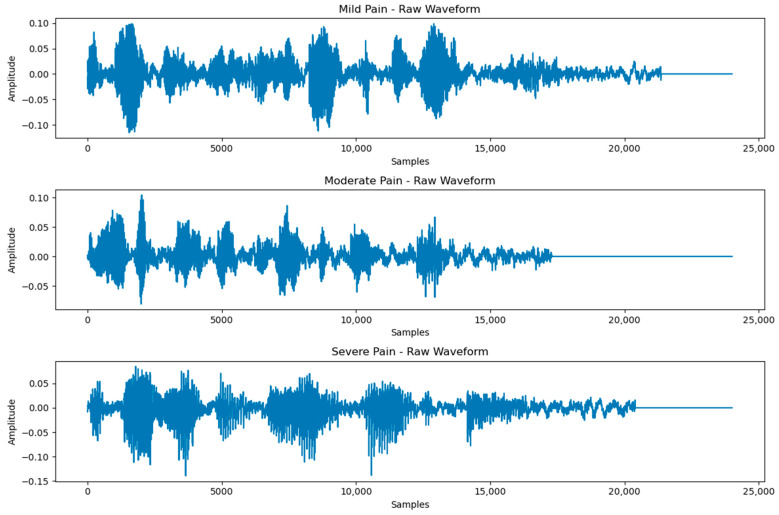
Raw speech waveforms for multiclass (mild, moderate, and severe levels) pain identification.

**Figure 5 diagnostics-15-02362-f005:**
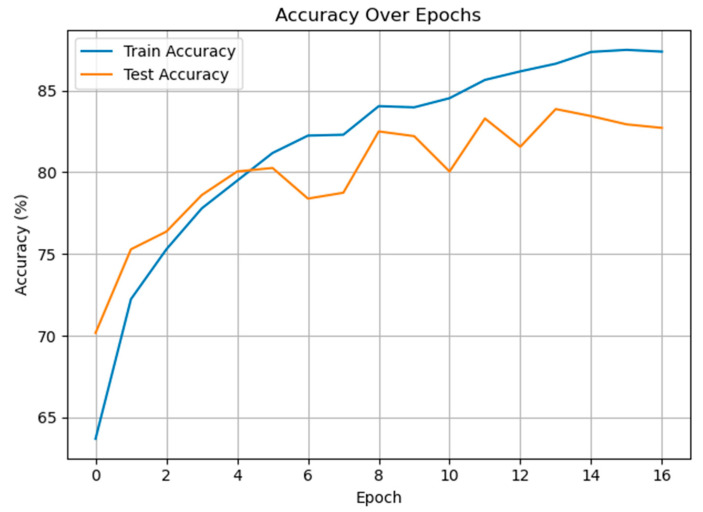
The training and test accuracies of the proposed method for pain vs. non-pain classification.

**Figure 6 diagnostics-15-02362-f006:**
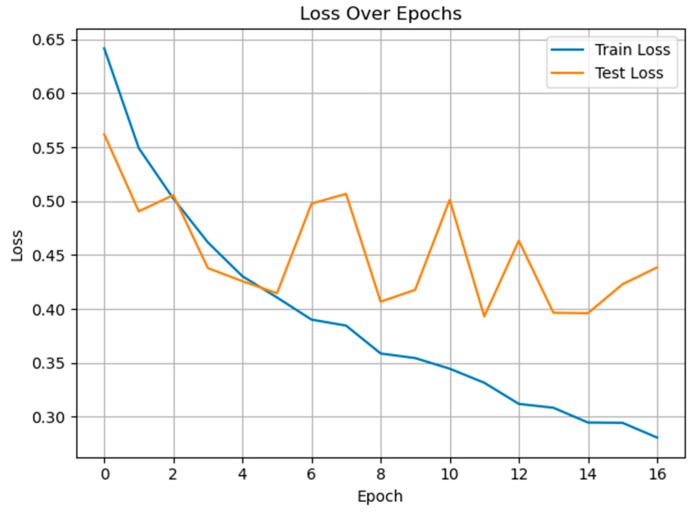
The training and test loss curves of the proposed method for pain vs. non-pain classification.

**Figure 7 diagnostics-15-02362-f007:**
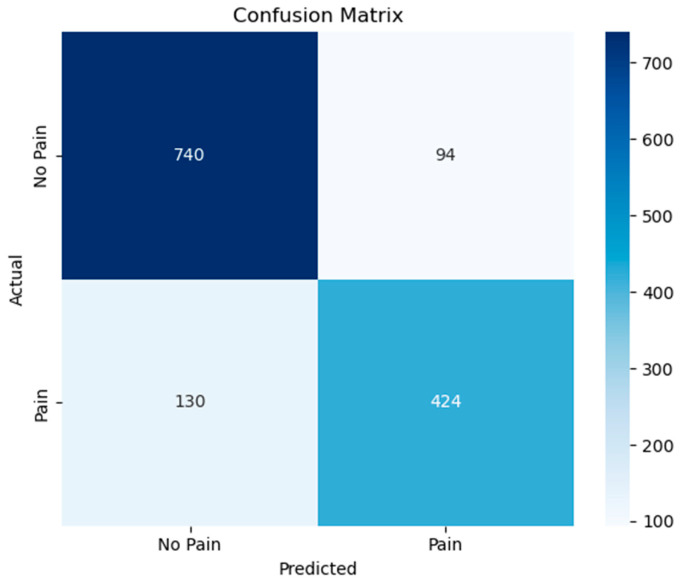
The confusion matrix for pain and non-pain classes.

**Figure 8 diagnostics-15-02362-f008:**
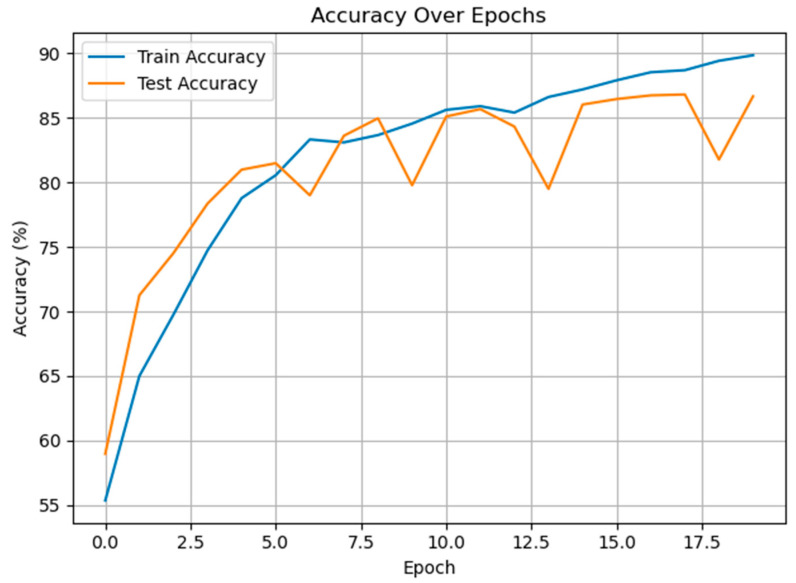
The training and test accuracies of the proposed method for cold vs. warm classification.

**Figure 9 diagnostics-15-02362-f009:**
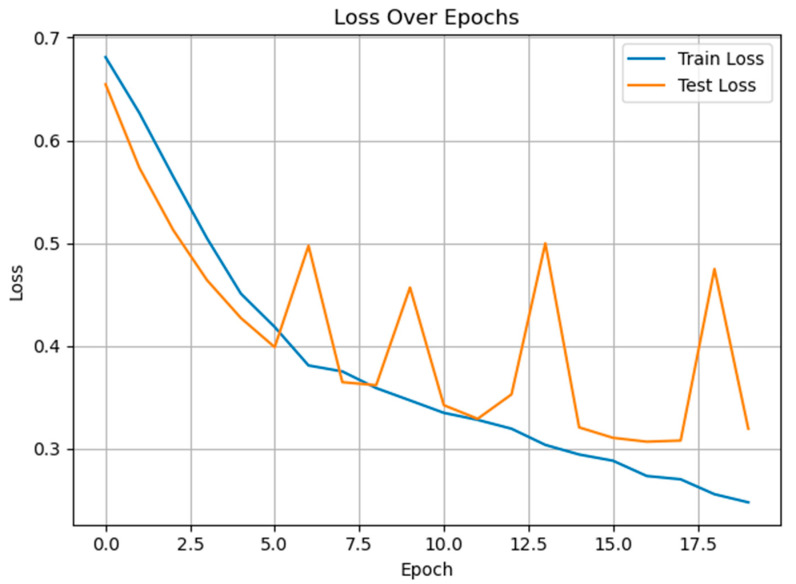
The training and test loss curves of the proposed method for cold vs. warm classification.

**Figure 10 diagnostics-15-02362-f010:**
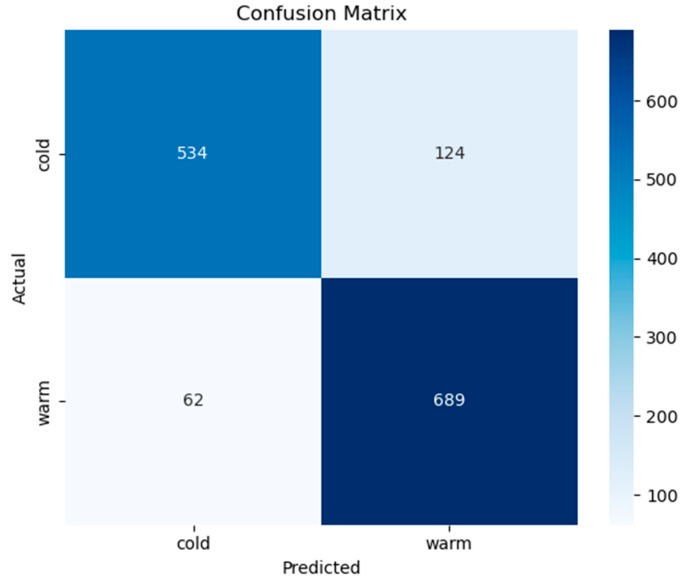
The confusion matrix for cold and warm classes.

**Figure 11 diagnostics-15-02362-f011:**
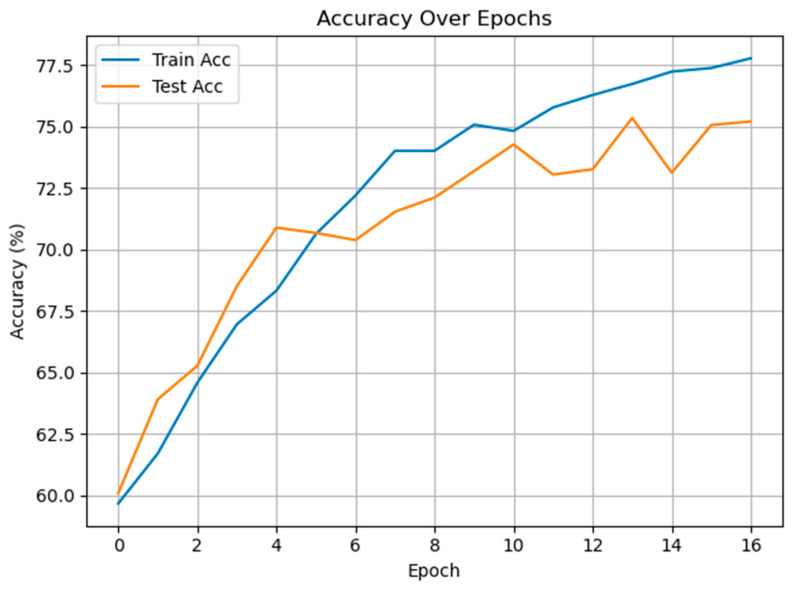
The training and test accuracies of the proposed method for multiclass pain classification.

**Figure 12 diagnostics-15-02362-f012:**
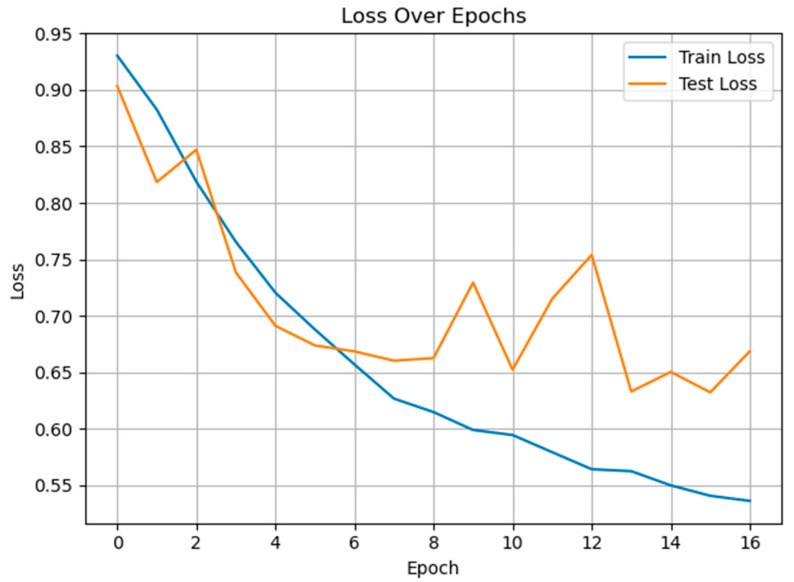
The training and test loss curves of the proposed method for multiclass pain classification.

**Figure 13 diagnostics-15-02362-f013:**
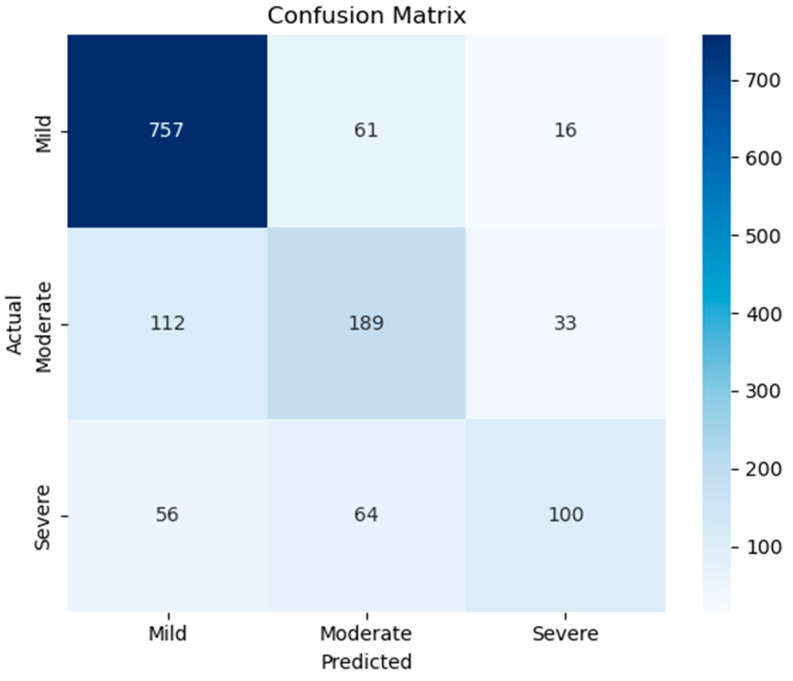
The confusion matrix for multiclass pain classes.

**Table 1 diagnostics-15-02362-t001:** Performance evaluation scores for pain and non-pain classification.

	Classes	Precision	Recall	F1-Score
**Class-Based Evaluation**	non-pain	0.8506	0.8873	0.8685
	pain	0.8185	0.7653	0.7910
**General Average** **Evaluation**	Macro Avg.	0.8346	0.8263	0.8298
	Weighted Avg.	0.8378	0.8386	0.8376
	Accuracy	0.8386		

**Table 2 diagnostics-15-02362-t002:** Performance evaluation scores for cold and warm condition.

	Classes	Precision	Recall	F1-Score
**Class-Based Evaluation**	cold	0.8960	0.8116	0.8517
	warm	0.8475	0.9174	0.8811
**General Average** **Evaluation**	Macro Avg.	0.8717	0.8645	0.8664
	Weighted Avg.	0.8701	0.8680	0.8673
	Accuracy	0.8680		

**Table 3 diagnostics-15-02362-t003:** Performance evaluation scores for multiclass (mild, moderate, and severe) pain level classification.

	Classes	Precision	Recall	F1-Score
**Class-Based Evaluation**	Mild	0.8184	0.9077	0.8607
	Moderate	0.6019	0.5659	0.5833
	Severe	0.6711	0.4545	0.5420
**General Average** **Evaluation**	Macro Avg.	0.6971	0.6427	0.6620
	Weighted Avg.	0.7430	0.7536	0.7435
	Accuracy	0.7536		

**Table 4 diagnostics-15-02362-t004:** Obtained results for ablation study.

Model	Accuracy	Precision	Recall	F1-Score
Full Model	0.8271	0.8257	0.7184	0.7683
No Dropout	0.8170	0.8378	0.6715	0.7455
No Weight Decay	0.8127	0.8467	0.6480	0.7342
No Early Stopping	0.8386	0.7957	0.8014	0.7986
No Dropout + No weight decay	0.8170	0.7187	0.8899	0.7952
GRU Only	0.8422	0.8587	0.7238	0.7855

**Table 5 diagnostics-15-02362-t005:** Performance comparison of the proposed method with various machine learning methods on multiclass pain level classification.

Model	Macro-Precision	Macro-Recall	Macro-F1-Score
LogReg	0.5509	0.5857	0.5533
SVM	0.5974	0.6363	0.6057
kNN-5	0.6297	0.5138	0.5378
DT	0.5483	0.5463	0.5473
RF	0.7633	0.5617	0.6006
GRU-Mixer	0.6971	0.6427	0.6620

## Data Availability

The original data presented in the study are openly available in TAME Pain: Trustworthy AssessMEnt of Pain from Speech and Audio for the Empowerment of Patients (version 1.0.0). PhysioNet. RRID:SCR_007345. https://doi.org/10.13026/20e2-1g10.

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
