# Peer review of "Pain Level Classification from Speech Using GRU-Mixer Architecture with Log-Mel Spectrogram Features"

_diagnostics, 2025, doi:10.3390/diagnostics15182362_

Round 1

Reviewer 1 Report

Comments and Suggestions for Authors

In this paper, a novel model called GRU-Mixer was developed and applied on speech based pain level classification. The paper was well written and organized. The experimental works were carried out accordingly. There are some typos,
1-    In the abstract: 
 ….. and efficiency in computation. Should be …. and computational efficiency. 
showcase….… with 83.86%. should be demonstrate ….. achieving 83.6%.
2-    In the introduction:
…which can be inacted or culture should be …which can be enacted…..
… both modalities, and should be ….both modalities and 
… SenseEmotion dataset, should be …..SenseEmotion dataset
…. trajectories, whereafter adaptive … should be …trajectories, where after adaptive …
3-    mel should be Mel
4-    Check the the fonts of Figure-1
5-    Make constant the writing of GRU-Mixer
6-    Why did the test loss function fluctuate? 
7-    Make consist of No-Pain or non-pain
8-    log should be Log

Author Response

In this paper, a novel model called GRU-Mixer was developed and applied on speech based pain level classification. The paper was well written and organized. The experimental works were carried out accordingly. There are some typos,
1-    In the abstract: 
 ….. and efficiency in computation. Should be …. and computational efficiency. 
showcase….… with 83.86%. should be demonstrate ….. achieving 83.6%.
2-    In the introduction:
…which can be inacted or culture should be …which can be enacted…..
… both modalities, and should be ….both modalities and 
… SenseEmotion dataset, should be …..SenseEmotion dataset
…. trajectories, whereafter adaptive … should be …trajectories, where after adaptive …
3-    mel should be Mel
4-    Check the the fonts of Figure-1

Answer: Yes, we checked it.
5-    Make constant the writing of GRU-Mixer

Answer: Yes, we did it.
6-    Why did the test loss function fluctuate? 

Answer:  Oscillations in loss curves reflect the trade-off between learning genuine pain-related acoustic patterns in the model and overfitting training data idiosyncrasies, scaled by dataset size, diversity, and stochastic training dynamics.
7-    Make consist of No-Pain or non-pain
8-    log should be Log

Answer: All mentioned typo errors are corrected.

Reviewer 2 Report

Comments and Suggestions for Authors

The authors are presenting the novel algorithm to recognize the pain from speech signals. Some remarks. It is very difficult to evaluate the efficiency of the proposed method. We can see that in other studies very different levels of accuracy has been achieved while the authors are using completely new database. In this situation it is hard to see how well proposed method works. Partially this problem could be solved performing some experiments with other databases too. 

Next issue is the selection of algorithm proposed by authors. In the current form we can get the impression that authors randomly selected some algorithm and are trying to apply it. 

Graphical material is not very informative: it is unclear what purpose serves signal oscilograms with pain and without pain; why so detailed information about the loss during training is necessary (fluctuations are largely random processes), etc. 

Author Response

Reviewer 2

The authors are presenting the novel algorithm to recognize the pain from speech signals. Some remarks. It is very difficult to evaluate the efficiency of the proposed method. We can see that in other studies very different levels of accuracy has been achieved while the authors are using completely new database. In this situation it is hard to see how well proposed method works. Partially this problem could be solved performing some experiments with other databases too. 

Answer: The following paragraphs are added to the end of the discussion section.

The proposed GRU-Mixer approach is demonstrated to possess clear novelty compared to existing work because it does not rely on hand-designed prosodic or spectral features but instead leverages the power of an end-to-end, fully trainable model. The architecture allows the model to simultaneously process both temporal and spectral features of pain speech as well as allowing multiple tasks such as binary pain detection, intensity estimation, and thermal condition classification. In this context, the framework offers a tunable alternative to traditional feature-based approaches.

In terms of effectiveness, reported performance is good, with around 84 percent accuracy for binary pain detection and 75 percent multiclass pain intensity classification. However, these are based only on the recently released TAME Pain dataset. Since there are no other published benchmarks yet for this corpus, and the previous research used to be based on smaller or less controlled datasets, it remains difficult to compare the performance of the proposed method to prior work directly. Extended experimental verification using various databases would help determine the system's generalizability.

One must also point out that, although the model shows excellent performance with binary classification tasks, the accuracy when recognizing subtler classes, such as moderate and severe pain, is lower. Furthermore, the current system relies exclusively on acoustic cues and lacks linguistic or multimodal data, which could improve performance in more realistic scenarios. Lastly, despite speaker-independent splits, the demographic and linguistic diversity of the dataset can, at worst, constrain how well the model can generalize to populations that are different.

Answer: We added the following to the discussion section:

To evaluate the efficiency of our proposed model, we did our best to compare it against several commonly used machine learning approaches on multiclass pain level classification. For this purpose, we included Logistic Regression (LogReg), Support Vector Machines (SVM), k-Nearest Neighbour with k set to 5 (kNN-5), Decision Tree (DT), and Random Forest (RF). Each method was configured with suitable parameters to ensure a fair comparison. LogReg was trained with a maximum of 2000 iterations and balanced class weights. SVM with linear kernel used C = 2.0, gamma set to “scale,” probability estimates enabled, and class weighting. kNN-5 was trained with k = 5. DT was trained with class weighting, while RF used 400 estimators and balanced subsample weighting. All models were trained on an 80/20 stratified train-test split to preserve class distribution, and their performance was assessed using macro precision, macro recall, and macro F1-score, respectively.

Table 5. Performance comparison of the proposed method with various machine learning methods on multiclass pain level classification

Model

Macro Precision

Macro Recall

Macro F1-score

LogReg

0.5509

0.5857

0.5533

SVM

0.5974

0.6363

0.6057

kNN-5

0.6297

0.5138

0.5378

DT

0.5483

0.5463

0.5473

RF

0.7633

0.5617

0.6006

GRU-Mixer

0.6971

0.6427

0.6620

Table 5 presents a comparison between the proposed GRU-Mixer model and several classical machine learning algorithms with macro precision, recall, and F1-score metrics, respectively. LogReg and DT received relatively lower scores, i.e., poor ability for maintaining performance on all the classes. SVM performed better overall, but more precisely in recall. The kNN-5 model possessed stronger precision but was marred by poor recall and hence a weaker F1-score. RF possessed the strongest precision among the traditional models but possessed weaker recall, which compromised its overall F1 performance. However, the proposed GRU-Mixer model provided a more balanced outcome, achieving higher macro recall than most baselines and a competitive macro precision, leading to the best macro F1-score across all methods.

Next issue is the selection of algorithm proposed by authors. In the current form we can get the impression that authors randomly selected some algorithm and are trying to apply it. 

Answer: The following is added to the discussion part:

The choice of the GRU-Mixer model might at first glance look as much a matter of random selection of a neural model for the task of pain recognition. The rationale originates in the particular strengths of recurrent networks for representing sequential information such as speech. Bidirectional GRU layers allow the model to capture both past and future temporal relationships in vocal signals, especially relevant for finding subtle prosodic and spectral content changes associated with pain. Incorporating temporal pooling also allows learning compact global representations without depending on handcrafted features. So, although the GRU-Mixer is quite lean compared to more complex structures, it is an informed compromise between expressiveness and computational expense, and therefore well-suited for real-time or low-resource clinical settings.

Graphical material is not very informative: it is unclear what purpose serves signal oscilograms with pain and without pain; why so detailed information about the loss during training is necessary (fluctuations are largely random processes), etc. 

Answer: We acknowledge the concern of the reviewer regarding the informativeness of the graphical material. The raw speech oscillograms for non-pain and pain conditions were included to provide a visual impression of the nature of the dataset and to highlight the diversity of the recorded signals. We acknowledge that these differences will not be explicitly evident to every reader, but we found them useful as a point of reference for future researchers who engage in research with the same dataset. For the exact training and test loss curves, our aim was to clarify the learning behavior of the proposed model and capture possible overfitting trends. We do like that test loss variation is partly random and can perhaps be managed with less extensive presentation. In future revisions, we will reduce the graphical content and emphasize more task-oriented figures, such as confusion matrices and performance plots, that more obviously describe the effectiveness of the method.

Round 2

Reviewer 2 Report

Comments and Suggestions for Authors

Most of the remarks has been taken into the account